# BCL11A Expression in Non-Small Cell Lung Cancers

**DOI:** 10.3390/ijms24129848

**Published:** 2023-06-07

**Authors:** Ewa Kątnik, Agnieszka Gomułkiewicz, Aleksandra Piotrowska, Jędrzej Grzegrzółka, Alicja Kmiecik, Katarzyna Ratajczak-Wielgomas, Anna Urbaniak, Natalia Glatzel-Plucińska, Piotr Błasiak, Piotr Dzięgiel

**Affiliations:** 1Division of Histology and Embryology, Department of Human Morphology and Embryology, Wroclaw Medical University, 50-368 Wroclaw, Poland; ewa.katnik@student.umw.edu.pl (E.K.); aleksandra.piotrowska@umw.edu.pl (A.P.); jedrzej.grzegrzolka@umw.edu.pl (J.G.); alicja.kmiecik@umw.edu.pl (A.K.); katarzyna.ratajczak-wielgomas@umw.edu.pl (K.R.-W.); natalia.glatzel-plucinska@umw.edu.pl (N.G.-P.); piotr.dziegiel@umw.edu.pl (P.D.); 2Department of Biochemistry and Molecular Biology, Wroclaw University of Environmental and Life Sciences, 50-375 Wroclaw, Poland; anna.urbaniak@upwr.edu.pl; 3Department and Clinic of Thoracic Surgery, Wroclaw Medical University, 53-439 Wroclaw, Poland; piotr.blasiak@umw.edu.pl; 4Lower Silesian Center of Oncology, Pulmonology and Hematology, 53-439 Wroclaw, Poland

**Keywords:** BCL11A, non-small cell lung cancer, Ki-67, slug, snail, twist

## Abstract

B-cell leukemia/lymphoma 11A (BCL11A) may be one of the potential biomarkers of non-small cell lung cancer (NSCLC). However, its role in the development of this cancer has not yet been precisely established. The aim of this study was to investigate the expression of BCL11A at the mRNA and protein levels in NSCLC cases and non-malignant lung tissue (NMLT) and to determine the relationship between BCL11A expression and the clinicopathological factors and Ki-67, Slug, Snail and Twist. The localization and the level of BCL11A protein were examined using immunohistochemistry (IHC) on 259 cases of NSCLC, and 116 NMLT samples were prepared as tissue microarrays and using immunofluorescence (IF) in the following cell lines: NCI-H1703, A549 and IMR-90. The mRNA expression of BCL11A was determined using real-time PCR in 33 NSCLC cases, 10 NMLT samples and the cell lines. BCL11A protein expression was significantly higher in NSCLC cases compared to NMLT. Nuclear expression was found in lung squamous cell carcinoma (SCC) cells, while cytoplasmic expression was demonstrated in adenocarcinoma (AC) cells. Nuclear expression of BCL11A decreased with increasing malignancy grade and correlated positively with Ki-67 and Slug and Twist expression. The opposite relationships were found for the cytoplasmic expression of BCL11A. Nuclear expression of BCL11A in NSCLC cells may affect tumor cell proliferation and change their phenotype, thus promoting tumor progression.

## 1. Introduction

Currently, lung cancer is the leading cause of cancer-related mortality in men, accounting for 21.5% of all cancers and ranks second after breast cancer in women (13.7%). In 2020, more than 2.2 million new cases and 1.8 million deaths from lung cancer were reported worldwide [1]. Lung cancer is one of the worst prognosis malignant tumors. The 5-year survival rate in the US is only 22% [2].

Smoking is the main risk factor for developing lung cancer. Approximately 80% of lung cancer cases are associated with long-term active and passive smoking [3,4]. Other risk factors include exposure to carcinogens (e.g., radon and asbestos), air pollutants, genetic mutations associated with genomic instability, age, infections with oncogenic viruses (e.g., the Epstein–Barr virus) and a history of cancer [3,4,5,6,7]. It has been demonstrated that genomic instability is related to gene mutations (including *EGFR*, *KRAS*, *TP53*, *ALK* and *CHEK1*) involved in cell signal transduction, proliferation, DNA damage repair and apoptosis. The action of the above factors leads to the accumulation of genetic and epigenetic changes, which may result in cancer development [7,8,9].

The main histological types of lung cancer are small-cell lung cancer (SCLC) and non-small cell lung cancer (NSCLC), which account for 20% and 80% of cases, respectively. In addition, NSCLC includes three subtypes: adenocarcinoma (AC; ~40%), squamous cell carcinoma (SCC; ~30%) and large-cell carcinoma (LCC; ~10%) [3]. In our study, we used SCC and AC, which are most frequently diagnosed. SCC occurs mainly in smokers and arises from bronchial epithelial cells, while AC is characteristic of non-smokers of both sexes and arises from type II pneumocytes [3].

B-cell leukemia/lymphoma 11A (BCL11A) belongs to a family of protein transcription factors containing C2H2 zinc finger domains [10]. There are five isoforms of BCL11A, which are formed by alternative pre-mRNA splicing. They differ in length, molecular weight and domain content, with the best studied being BCL11A-XL, which is the longest isoform [6,10,11,12]. Abnormal translocations of the *BCL11A* locus [t(2,14)(p16;q32.3)] were associated with the development of chronic B-cell lymphocytic leukemia [10,13,14]. These findings contributed to further studies on hematological cancers, which proved the involvement of BCL11A in carcinogenesis [14,15,16,17,18].

Many other important functions of this protein have also been reported. BCL11A is a transcriptional regulator of many genes involved in normal cell function, including proliferation and apoptosis. BCL11A interacts with the anti-apoptotic proteins (BCL2 and BCL-XL) and the negative regulators of p53 (MDM2, MDM4 and SIRT1) [19,20].

In addition, studies on Burkitt lymphoma (EB1) cell lines and diffuse large B-cell lymphoma (SU-DHL-6) cell lines found that siRNA-mediated inhibition of *BCL11A* expression reduced cell viability and directed cells toward apoptosis [15]. This suggests that BCL11A may have an anti-apoptotic effect, thus influencing carcinogenesis. Elevated expression levels of BCL11A have been reported in many cancers, including breast cancer, prostate cancer, colorectal cancer and B-cell lymphomas and leukemias [17,21,22,23,24,25]. Higher levels of BCL11A were associated with tumor growth and shorter survival [17,21,22,23]. In addition, Nakamura et al. demonstrated that high BCL11A expression led to neoplastic transformation of the normal mouse fibroblast cell line (NIH/3T3) [11].

Although many studies have demonstrated the involvement of BCL11A in carcinogenesis, the role of this protein in NSCLC has not been confirmed yet. Jiang et al. found that BCL11A mRNA expression was three-fold higher in NSCLC than in normal lung tissue and correlated with smoking and the histologic subtype of SCC [8]. Moreover, high expression of this protein was associated with a better prognosis, especially in early-stage SCC [8]. These findings were partially confirmed by Boelens et al. [26]. Similar results were obtained by Zhang et al., who studied the longest isoform of the protein, BCL11A-XL. They showed that high levels of this isoform were associated with longer disease-free survival in SCC and LCC patients [6]. In turn, the opposite results were obtained by Liao et al., who noted that high BCL11A expression was associated with a worse prognosis [27].

The exact effect of BCL11A on the development and progression of NSCLC remains largely unknown. So far, BCL11A is considered one of the proteins included in the mammalian chromatin remodeling complex (BAF). The function of the BAF complex is the regulation of cell proliferation and differentiation, DNA repair and maintaining chromosome stability. Therefore, the presence of mutations in genes encoding BAF proteins, e.g., *BCL11A*, is closely related to genomic instability in lung cancer, regardless of the subtype [28]. Moreover, genomic instability is associated with faster disease recurrence and increased mortality in patients with NSCLC [7]. These findings may confirm the role of BCL11A in the development and progression of NSCLC. Subsequent studies have shown that silencing *BCL11A* inhibits the growth of SCC [29]. Currently, four ways of regulating BCL11A in lung cancer are known: by miRNA—miR-30a (silencing), amplification (activation), long non-coding RNA—DSCAM-AS1 (activation) and SOX2 (regulation at the transcription level) [8,27,29].

Due to the inconclusive reports, the aim of our study was to examine BCL11A expression at the mRNA and protein levels in NSCLC, non-malignant lung tissue (NMLT) and NSCLC cell lines. In addition, we determined the relationship between BCL11A expression levels and the clinicopathological factors, Ki-67 and epithelial–mesenchymal transition (EMT) markers, i.e., Slug, Snail and Twist.

## 2. Results

### 2.1. BCL11A Protein Expression in NSCLC, NMLT and Cell Lines

In total, 214 cases (82.6%) of 259 cases of NSCLC showed BCL11A expression. However, the expression site differed depending on the tumor subtype. Nuclear expression was found in 66 of 96 cases of SCC (68.8%; Figure 1B), but cytoplasmic expression was only reported in 20 cases. As regards AC, a cytoplasmic reaction was observed in 120 cases (73.6%; Figure 1C) and a nuclear reaction in 8 cases.

The mean value of nuclear expression BCL11A in SCC was 1.500 ± 1.238 based on a 3-point scale. Low nuclear expression was observed in 30 SCCs (31.3%), while high expression was reported in 66 cases (68.7%). Based on a 12-point scale, the mean value of the cytoplasmic expression of BCL11A in AC was 2.291 ± 2.365. Low cytoplasmic expression was demonstrated in 76 AC cases (46.6%), while high expression was observed in 87 cases (53.4%). In addition, lymphocytes forming inflammatory infiltrates showed nuclear expression. Low cytoplasmic expression of BCL11A was found in 31.9% of NMLT specimens, while 68.1% of cases showed no expression of the protein (Figure 1A).

A statistical analysis showed that BCL11A nuclear and cytoplasmic expression levels were significantly higher in NSCLC cells compared to NMLT (*p* < 0.0001 in both cases). Further relationships were observed when the entire NSCLC cohort was divided into SCC and AC subtypes. The nuclear expression of BCL11A was significantly higher in SCC (1.500 ± 1.238) and AC (0.055 ± 0.255) compared to NMLT (*p* < 0.0001, *p* = 0.016, respectively; Figure 2A).

Cytoplasmic expression of the protein was also significantly higher in AC (2.291 ± 2.365) compared to NMLT (0.397 ± 0.658; *p* < 0.0001; Figure 2B). Moreover, the level of BCL11A differed significantly between SCC and AC in nuclear (*p* < 0.0001) and cytoplasmic expression (*p* < 0.0001). In addition, we showed an inverse correlation between BCL11A expression levels in the nucleus and in the cytoplasm, i.e., an increase in nuclear expression was associated with a significant decrease in cytoplasmic expression (r = −0.487, *p* < 0.0001; Figure 2C).

BCL11A protein levels were also higher in the NSCLC cell lines (NCI-H1703 and A549) compared to the control cells (IMR-90). The results were confirmed using IF (Figure 3A–C).

In addition, the NCI-H1703 SCC cell line showed nuclear expression (Figure 3B), and the A549 (AC) line showed cytoplasmic expression of the protein (Figure 3C). However, in the IMR-90 normal lung fibroblast cell line, BCL11A was localized mainly in the cytoplasm (Figure 3A).

### 2.2. Relationship between BCL11A Protein Expression and the Clinicopathological Factors in NSCLC

In our study, we showed a decrease in the nuclear expression of BCL11A with increasing histological grade (G) of NSCLC. The mean expression value was significantly higher in cases with G1 and G2 (0.772 ± 1.169) compared to G3 (0.250 ± 0.641; *p* = 0.0003; Figure 4A, Table 1).

However, we did not observe significant relationships between the nuclear expression of BCL11A and histological grade in SCC and AC cases that were analyzed separately (Table 1).

Moreover, the results obtained from the entire NSCLC cohort showed a weak correlation of nuclear BCL11A expression with the primary tumor size measured in centimeters (r = 0.127, * *p* = 0.0455; Figure 4C). Tumors < 4 cm showed lower expression of the protein (0.536 ± 1.033) compared to tumors of larger sizes (0.679 ± 1.065; Table 1). The opposite trend was noted when only SCCs were analyzed, i.e., smaller tumors were characterized by higher BCL11A expression (1.795 ± 1.196 vs. 1.268 ± 1.225; *p* = 0.0422; Table 1).

The analysis of the results of all NSCLC cases also showed a positive correlation of the nuclear expression of BCL11A with the percentage of necrosis (r = 0.233, *** *p* = 0.0006; Figure 4E). The mean expression of BCL11A was significantly higher when the percentage of necrosis was ≥32% (1.036 ± 1.253) compared to cases with the percentage of necrosis <32% (0.417 ± 0.867; Table 1).

In addition, when we analyzed AC cases, we noted higher nuclear expression of BCL11A in pT3 and pT4 tumors compared to pT1 and pT2 (*p* = 0.0134) and in cases with tumor stage III and IV compared to I and II (*p* = 0.0341; Table 1).

We showed no statistically significant relationships of the nuclear expression of BCL11A with sex, age or lymph node metastasis status (N) when the entire cohort and the individual lung cancer subtypes were assessed (Table 1).

Interestingly, we obtained opposite results in the case of the cytoplasmic expression of BCL11A. The cytoplasmic expression levels of BCL11A were significantly higher in cases with histological grade G3 (1.844 ± 2.137) compared to G1 and G2 (1.470 ± 2.197; *p* = 0.0325; Figure 4B, Table 2).

We found no statistically significant differences in the cytoplasmic expression of BCL11A in relation to the histological grade of SCC and AC subtypes (Table 2). In addition, we found a negative correlation of the cytoplasmic expression of BCL11A with the size of the primary tumor measured in centimeters (r = −0.228, *p* = 0.0001; Figure 4D) and the percentage of necrosis (r = −0.299, *p* < 0.0001; Figure 4F). We obtained similar results when only AC cases were analyzed (*p* = 0.0501 and *p* = 0.0246, respectively; Table 2).

The analysis of the entire cohort and individual NSCLC subtypes showed no statistically significant differences in the cytoplasmic expression of BCL11A in relation to sex, age, primary tumor size (T), lymph node metastasis status (N) and cancer stage (Table 2).

### 2.3. BCL11A mRNA Expression in NSCLC, NMLT and Cell Lines

The results showed higher BCL11A mRNA expression in NSCLC (3.673 ± 6.918) compared to NMLT (1.905 ± 1.644). In addition, significantly higher BCL11A mRNA expression levels were reported in SCC (15.066 ± 7.586) than in AC (3.819 ± 3.499; *p* = 0.002; Figure 5A).

The analysis of the correlations between BCL11A mRNA expression and the clinicopathological factors in the entire NSCLC cohort showed that higher BCL11A mRNA expression was associated with smaller primary tumors (pT1 and pT2; *p* = 0.0255). However, no significant correlations were found between BCL11A mRNA expression and other clinicopathological factors when all NSCLC cases and individual histological subtypes were considered.

In vitro studies showed higher BCL11A mRNA expression levels in NSCLC cell lines (NCI-H1703 and A549) compared to the normal lung fibroblast line IMR-90. Higher levels of BCL11A mRNA expression were found in NCI-H1703 (lung SCC) compared to A549 (AC) (13.989 ± 2.963 and 5.841 ± 0.604; *p* = 0.0095, respectively) and the normal lung fibroblast line IMR-90 (1.391 ± 0.717; *p* = 0.002; Figure 5B). In addition, there were statistically significant differences in BCL11A mRNA expression between the A549 and IMR-90 cell lines (*p* = 0.0012).

### 2.4. Relationship between BCL11A Expression and Tumor Cell Proliferation and Epithelial–Mesenchymal Transition

The nuclear expression of BCL11A in NSCLC cells correlated positively with Ki-67 expression (r = 0.42, *p* < 0.0001; Figure 6A). However, the cytoplasmic expression of BCL11A correlated negatively with Ki-67 expression (r = −0.28, *p* < 0.0001; Figure 6B).

In NSCLC cases, we observed the nuclear and cytoplasmic expression of Slug and Twist, as well as the cytoplasmic expression of Snail. We demonstrated a statistically significant positive correlation of the nuclear expression of BCL11A with the nuclear expression of Slug (r = 0.66, *p* < 0.0001; Figure 6C) and Twist (r = 0.47, *p* < 0.0001; Figure 6G). A similar trend was observed when we analyzed the nuclear expression of BCL11A and cytoplasmic expression of Snail. However, the relationship was not statistically significant (r = 0.13, *p* = 0.0251). However, a significant negative correlation was noted between the nuclear expression of BCL11A and Slug expression in the cytoplasm (r = −0.31, *p* < 0.0001; Figure 6E).

The cytoplasmic expression of BCL11A correlated negatively with the nuclear expression of Slug (r = −0.42, *p* < 0.0001; Figure 6D) and Twist (r = −0.21, *p* = 0.0003; Figure 6H) and positively with the cytoplasmic expression of Slug (r = 0.26, *p* < 0.0001; Figure 6F). However, no statistically significant relationship was found between the cytoplasmic expression of BCL11A and the cytoplasmic expression of Twist (r = 0.16, *p* = 0.0057) and Snail (r = −0.16, *p* = 0.005).

### 2.5. Relationship between BCL11A Expression and Overall Survival

A log-rank analysis showed no significant relationship between BCL11A expression levels and the overall survival of patients with NSCLC. Patients with higher nuclear expression of BCL11A showed a slightly longer survival compared to those with low nuclear BCL11A levels. However, these differences were not statistically significant (*p* = 0.1850; Figure 7A). As regards the cytoplasmic expression of BCL11A, no relationship was found between the protein level and survival (*p* = 0.5019; Figure 7B).

In addition, the univariate and multivariable Cox proportional hazard analysis was performed in the entire group of NSCLC patients (Table 3) and separately for patients with SCC (Table 4) and AC (Table 5). The univariate analysis showed that higher nuclear expression levels of BCL11A were associated with longer survival in NSCLC patients (*p* = 0.0360; Table 3). However, this relationship was not confirmed by the multivariate analysis (*p* = 0.0682; Table 3). We found no relationship between higher nuclear expression of BCL11A and a better prognosis when SCC and AC cases were analyzed separately. Additionally, no statistically significant differences were found for the cytoplasmic expression of BCL11A.

## 3. Discussion

Lung cancer is one of the most prevalent cancers worldwide with one of the worst prognosis. In 2020 alone, almost 1.8 million deaths due to lung cancer were reported worldwide [1]. Despite various treatment modes, new treatment modalities are sought to increase survival rates. Recently, much hope has been placed on identifying prognostic and predictive markers to facilitate the prognostic evaluation and treatment of lung cancer. BCL11A can be one of such factors. To the best of our knowledge, our study is the first to differentiate the expression levels of BCL11A due to its localization in NSCLC cells, NMLT and NSCLC cell lines. Lung SCCs were characterized using nuclear expression, while ACs were characterized using cytoplasmic expression. A similar relationship was noted by Jiang et al. However, they used a single scale when assessing nuclear and cytoplasmic reactions, hence the uniformity of the results [8]. Our clinical findings were confirmed by our in vitro studies as the SCC cell line (NCI-H1703) showed nuclear expression, while the AC cell line (A549) showed cytoplasmic expression of BCL11A. Moreover, the statistical analysis showed that the nuclear and cytoplasmic expression levels of BCL11A were significantly higher in NSCLC cells than in normal tissue. Similar results were obtained when SCC and AC subtypes were considered. These results are consistent with the findings of Liao et al. and Jiang et al. [8,27] and suggest the involvement of BCL11A in the development of NSCLC.

The nuclear localization of BCL11A is most often observed as this protein is a transcription factor. In the cell nucleus, it regulates the transcription of many genes related to apoptosis, e.g., *BCL2*, *BCL-XL* and *MDM2* or cell cycle control (e.g., *p21*) [20]. It was suggested that BCL11A was involved in inhibiting apoptosis [20].

Furthermore, we obtained similar results at the mRNA level. We demonstrated higher mRNA expression of BCL11A in NSCLC compared to NMLT. In addition, we showed significantly higher BCL11A mRNA expression levels in SCC than in AC, which is in line with Lazarus et al. who identified *BCL11A* as an oncogene of SCC due to its overexpression in SCC [29]. According to them, it was related to the amplification of locus (*BCL11A*) and regulation by SOX2. Additionally, silencing of *BCL11A* reduced the levels of SCC markers (KRT5 and TP63) and inhibited tumor growth [29]. Similar results were obtained by Zhang et al., although they conducted studies on one of the isoforms (BCL11A-XL). They found the elevated expression of BCL11A in NSCLC, particularly in SCC [6]. The overexpression of BCL11A at the mRNA and protein levels was also demonstrated in B-cell lymphoma, B-cell chronic lymphocytic leukemia and triple-negative breast cancer [15,21,30].

Our in vitro study with cell lines confirmed the results obtained in the clinical samples. Higher levels of BCL11A mRNA expression were found in NSCLC cell lines (NCI-H1703 and A549) compared to the normal lung fibroblast cell line (IMR-90). The lung SCC cell line (NCI-H1703) showed a higher level of BCL11A mRNA expression than the AC cell line (A549) and the normal lung fibroblast cell line (IMR-90). These results are consistent with the findings of Lazarus et al. [29]. In addition, we demonstrated higher levels of BCL11A in NSCLC cell lines compared to control cells. *BCL11A* is one of the genes encoding the protein that is part of BAF complexes (mammalian chromatin remodeling complexes), which are involved in the regulation of transcription, cell proliferation, DNA repair and maintenance of chromosomal stability [28]. Mutations in these genes resulted in genomic instability and the development of cancers, including lung cancer [28,31]. This confirms the importance of *BCL11A* overexpression in the initiation of NSCLC.

Additionally, we showed that the nuclear expression of BCL11A decreased with increasing histological grade in NSCLC. In turn, we observed a positive correlation between the nuclear expression of the protein and the size of the primary tumor (in cm) and the percentage of necrosis. Interestingly, when SCC was analyzed separately, we found that smaller tumors showed significantly higher nuclear expression of BCL11A. These findings suggest that the downregulation of the nuclear expression of BCL11A may be crucial for tumor growth, especially in SCC. Jiang et al. also noted the relationship of the high expression of BCL11A with the absence of lymph node metastasis and lower disease stage [8]. This was confirmed by Boelens et al., who found that higher BCL11A expression levels were related to cases without metastases [26]. In our study, we observed a similar trend when the nuclear expression of BCL11A was analyzed. However, not all relationships were statistically significant. As regards the cytoplasmic expression of BCL11A, we obtained opposite results, i.e., the level of the protein in the cytoplasm of NSCLC cells increased with increasing histological grade. Moreover, in the whole cohort and the AC subtype, we observed a decrease in the cytoplasmic expression of BCL11A with increasing tumor size and the percentage of necrosis.

In the context of the above results, we found an interesting correlation, i.e., an increase in the nuclear expression of BCL11A was associated with a significant decrease in its cytoplasmic expression. This perhaps demonstrates the existence of a complex regulation of BCL11A transport from the cytoplasm to the cell nucleus (where the protein functions as a transcription factor) and from the cell nucleus to the cytoplasm (where the function of the protein is altered). However, the mechanisms of this process and the factors that would trigger it are unknown.

In our study, however, we observed some tendency. A decrease in nuclear expression was reported with the increase in histological grade, which shows that the type of BCL11A expression may be crucial for the further condition of the patient. However, further studies are warranted to confirm this relationship and to establish the significance of the presence of BCL11A in the cytoplasm.

To date, various mechanisms of the regulation of *BCL11A* gene expression have been described through miRNA or the amplification of locus (*BCL11A*). It was found that miR-30a could inhibit BCL11A protein expression in vitro. However, miR-30a was downregulated in NSCLC tissues [8]. On the other hand, Lulli et al. proved that the inhibition of miR-486-3p in pancreatic and lung cancer cells promoted tumor growth by increasing BCL11A levels [32]. In addition, the gene amplifications much more frequently observed in NSCLC increased BCL11A expression [8]. The diverse modes of regulation and the occurrence of the post-transcriptional modifications of BCL11A pre-mRNA suggest a very complex way of controlling BCL11A expression levels, which is most likely dependent on the type of cancer and possibly on the histological type of the same tumor.

In addition, our study is the first in which BCL11A expression levels were correlated with the cell proliferation marker Ki-67 and the epithelial–mesenchymal transition (EMT) markers such as Slug, Snail and Twist. So far, similar studies have been conducted only on breast cancer samples. Shen et al. showed no relationship between BCL11A expression and Ki-67 [33]. In turn, Zhu et al. found that BCL11A overexpression promoted tumor cell migration and EMT through the activation of the Wnt/β-catenin pathway, which interacts with Snail [34]. In addition, high expression of BCL11A was associated with the overexpression of Slug and Snail [34].

Our study demonstrated that the localization of BCL11A in the cell may be important for its effects on the proliferation and EMT of NSCLC cells. The nuclear expression of BCL11A correlated positively with Ki-67 antigen expression, which suggests that the presence of the protein in the cell nucleus may stimulate cell division. As regards the cytoplasmic expression of BCL11A, an inverse relationship was observed with Ki-67 expression, which confirms the role of BCL11A in proliferation. We also showed positive correlations between the nuclear expression of BCL11A and the nuclear expression of Slug and Twist. However, a negative correlation was found with the cytoplasmic expression of Slug. As regards the cytoplasmic expression of BCL11A, we obtained opposite results, which confirms the significance of the localization of the protein in the cell as a transcription factor. Probably BCL11A has a fairly significant role in the activation of cell proliferation and EMT of NSCLC cells, which is in line with the results obtained from breast cancer samples by Zhu et al. [34]. In turn, Zhang et al. found that the downregulation of BCL11A expression in prostate cancer cells resulted in the suppression of proliferation and EMT [22]. These results show the association between BCL11A expression levels and NSCLC cell proliferation and EMT.

In our study, we did not demonstrate the prognostic significance of BCL11A expression in patients with NSCLC. However, we observed a trend indicating that higher nuclear expression of the protein could be associated with slightly longer overall survival. This is consistent with other studies in which high expression of BCL11A correlated with longer overall survival and disease-free survival in NSCLC, especially in patients with SCC [8]. Additionally, the loss of expression of the BCL11A-XL isoform was a negative prognostic factor in NSCLC and diffuse large B-cell lymphomas (DLBCLs) [6,35]. However, we did not demonstrate any association between the cytoplasmic expression of BCL11A and the overall survival of patients with NSCLC.

Unfortunately, our study is not free from certain limitations. One of them is that it is a single-center study, covering only patients from the Polish population. Therefore, the results still need to be validated in a multicenter study including a diverse group of patients. In addition, BCL11A mRNA expression studies were performed on a limited number of NSCLC cases because we had to exclude samples with lymphocytic infiltration. Lymphocytes show a high expression of BCL11A [20]. To avoid false BCL11A mRNA expression results due to the presence of lymphocytes among the cancer cells, we had to significantly reduce the number of samples for molecular studies.

In conclusion, determining the expression level of BCL11A, its localization in cancer cells and relationship with clinicopathological factors may bring us closer to the understanding of the role of this protein in carcinogenesis. Our findings suggest that the nuclear expression of BCL11A in NSCLC cells may affect the proliferation of tumor cells and change their phenotype, thus promoting tumor progression. However, we still do not know the exact mechanisms of BCL11A, its interactions with other proteins, or the signaling pathways in which BCL11A is involved and through which it affects normal and cancer cells. Further functional studies are needed to better understand the biology and function of this protein.

## 4. Materials and Methods

### 4.1. Patients

The study material consisted of 259 cases of NSCLC and 116 samples of NMLT taken from patients who underwent surgery at the Lower Silesian Center of Lung Diseases in Wroclaw between 2007 and 2016. All study samples were collected before chemotherapy and radiotherapy administration. The study group included 96 SCCs and 163 ACs. NMLT samples obtained from tumor margins were used as controls. A formalin-fixed, paraffin-embedded (FFPE) technique was used for all samples. In addition, NSCLC cases (21 SCCs, 12 ACs) and NMLT (*n* = 10) as controls, which were fixed in RNA*later* (Thermo Fisher Scientific, Waltham, MA, USA), were used for polymerase chain reaction (PCR). Histological subtypes and malignancy grades (Gs) of NSCLC were evaluated in hematoxylin- and eosin-stained slides by two independent histopathologists based on the World Health Organization (WHO) classification [36]. To confirm the histological subtype of NSCLC, immunohistochemical reactions anti-p63 (for SCC) and anti-TTF-1 (for AC) were performed. The TNM classification was based on the recommendations of the International Association for the Study of Lung Cancer (IASLC) [37]. Clinicopathological data of each case are given in Table 6 and Table 7. The study was approved by the Bioethics Committee at Wroclaw Medical University (No. KB 318/2018 of 28 May 2018).

### 4.2. Cell Lines

BCL11A expression was examined in two NSCLC cell lines [NCI-H1703 (ATCC, Manassas, VA, USA; CRL-5889), A549 (ATCC; CCL-185)] and normal lung fibroblast cell line (IMR-90) (ATCC; CCL-186). NCI-H1703 was derived from stage I SCC, while A549 was the AC cell line. IMR-90 was cultured in Minimum Essential Medium supplemented with non-essential amino acids (Sigma-Aldrich, Saint Louis, MO, USA). RPMI-1640 (Lonza, Basel, Switzerland) was used to culture NCI-H1703, while A549 was grown in Kaighn’s Modification of Ham’s F12 Medium (ATCC). All culture media were supplemented with 10% fetal bovine serum, L-glutamine (2 mM) and a 1% mixture of antibiotics (100 U/mL penicillin G and 100 µg/mL streptomycin). The reagents were obtained from Sigma-Aldrich. The cells were grown under constant conditions (37 °C in an incubator, 5% CO_2_, 95% humidity) (HERAcell 150i, Thermo Fisher).

### 4.3. Tissue Microarrays (TMAs)

Tissue microarrays (TMAs) were performed from 259 NSCLC and 116 NMLT sections using an automated TMA Grand Master (3DHistech, Budapest, Hungary) according to the manufacturer’s instructions. All tumors were reviewed by two pathologist researchers (P.D., A.P.). For each case, the histopathologist selected three representative spots with a diameter of 1.5 mm from an archival paraffin block based on the previously prepared slides stained with hematoxylin and eosin. The specimens were scanned using the Panoramic MIDI II histology scanner (3DHistech).

### 4.4. Immunohistochemistry (IHC)

Immunohistochemical reactions were performed on 4 µm TMA sections prepared on Superfrost Plus slides (Menzel Gläser, Braunschweig, Germany) using the DAKO Autostainer Link48 (Dako, Glostrup, Denmark), which resulted in maintaining constant conditions for all cases. Initially, the specimens were deparaffinized and hydrated, and epitopes were exposed by boiling at 97 °C for 20 min in PT-Link in EnVision^TM^ FLEX Target Retrieval Solution (pH 6.0 for Ki-67 antibodies, pH 9.0 for BCL11A, Slug, Snail and Twist). Endogenous peroxidase was blocked using EnVision^TM^ FLEX Peroxidase-Blocking Reagent (5 min) at room temperature (RT). EnVision^TM^ FLEX was used to visualize IHC reactions according to the manufacturer’s instructions. For the IHC reactions, we used primary antibodies directed against BCL11A (mouse, monoclonal, Abcam, Cambridge, UK; ab19489, 18B12DE6; 1:200, 20 min, at RT), Slug (mouse, monoclonal, Santa Cruz Biotechnology, Santa Cruz, CA, USA; sc-166476, 1:50, 20 min, at RT), Snail (rabbit, polyclonal, ProteinTech, Rosemont, Illinois, USA; 13099-1-AP, 1:400, 20 min, at RT), Twist (mouse, monoclonal, Abcam; ab50887, 1:50, 20 min, at RT) and Ki-67 (mouse, monoclonal, MIB-1 clone, Dako; IS626; ready-to-use, 20 min, at RT) diluted in FLEX Antibody Diluent. As regards Twist staining, slides were additionally incubated with EnVision FLEX+ Mouse LINKER for 15 min at RT to enhance the signal. The slides were overlaid with secondary antibodies conjugated with EnVision^TM^ FLEX/horseradish peroxidase (HRP) for 20 min at RT. To obtain the color reaction, the sections were incubated with the substrate for peroxidase 3,3′-diaminobenzidine (DAB) for 10 min at RT. The preparations were counterstained with hematoxylin (EnVision^TM^ FLEX Hematoxylin) and dehydrated in ethanol solutions with increasing concentrations (70%, 96%, 99.9%) and xylene. Between each step, the preparations were washed in EnVision^TM^ FLEX Wash Buffer. With the exception of some antibodies, the reagents and equipment were obtained from Dako.

### 4.5. Evaluation of Immunohistochemical Reactions

Evaluation of the intensity of BCL11A expression was conducted by two independent investigators (P.D., E.K.) using a BX41 light microscope (Olympus, Tokyo, Japan). Assessment of IHC reactions was performed at ×200 magnification. Three representative spots of 1.5 mm in diameter were evaluated for each case. Each spot contained an average of 4000–6000 cells. The cytoplasmic reaction was determined using the 12-point semi-quantitative immunoreactive score (IRS), according to Remmele and Stegner [38]. The scale is the product of two variables: the percentage of positive cells and the intensity of the color reaction (A × B, Table 8).

A point scale was used to evaluate nuclear reaction taking into account the percentage of positive cells: 0 points ≤ 5%, 1 point—6–25%, 2 points—26–50% and 3 points > 50% positive cells (Table 9) [39].

### 4.6. RNA Isolation and cDNA Synthesis

Total RNA was isolated from NSCLC cases, NMLT and cell lines (IMR-90, NCI-H1703, A549) using the RNeasy Mini Kit (Qiagen, Hilden, Germany) according to the manufacturer’s instructions. A NanoDrop1000 spectrophotometer (Thermo Fisher) was used to measure the concentration and purity of RNA samples. Reverse transcription reactions were performed with 500 ng of total RNA using the High Capacity cDNA Reverse Transcription Kit (Applied Biosystems, Foster City, CA, USA). The reactions were performed under the following conditions: 10 min at 25 °C, 120 min at 37 °C and 5 min at 85 °C in a SimpliAmp Thermal Cycler (Applied Biosystems).

### 4.7. Real-Time PCR

BCL11A mRNA levels in NSCLC cases, NMLT and the selected cell lines (IMR-90, NCI-H1703, A549) were examined using real-time PCR using 7500 Real-Time PCR System and TaqMan Gene Expression Master Mix (Applied Biosystems). *β-actin* was the reference gene to normalize the results. For the reactions, the following sets of primers and TaqMan probes were used: Hs00256254_m1 for *BCL11A* and Hs99999903_m1 for *β-actin* (Applied Biosystems). The reactions were performed in triplicates under the following conditions: polymerase activation at 50 °C for 2 min, initial denaturation at 95 °C for 10 min, followed by 45 cycles involving denaturation at 95 °C for 15 s and primer attachment and DNA synthesis at 60 °C for 1 min. The relative expression level of BCL11A mRNA was calculated using the ΔΔCt method.

### 4.8. Immunofluorescence (IF)

Immunofluorescence (IF) was used to examine the location and expression levels of BCL11A in the cell lines (IMR-90, NCI-H1703, A549). Initially, 2 × 10^4^ cells were seeded into each well of the Millicell EZ 8-well glass slide (Merck, Darmstadt, Germany; PEZGS0816). After 48 h of culture, the specimens were fixed with acetone/methanol (1:1) at 4 °C for 15 min. Next, 3% BSA in 0.4% TBST buffer blocked non-specific binding sites (45 min at RT). The specimens were incubated overnight with primary mouse anti-BCL11A monoclonal antibodies (ab19489, Abcam; 1:200) in Antibody Diluent (Dako) at 4 °C. The negative control was incubated only from PBS instead of the specific antibody. Next, secondary antibodies (Alexa Fluor 488; ab150113, Abcam; 1:6000) were used for 1 h at RT. The specimens were enclosed in Fluoroshield Mounting Medium with DAPI (Abcam; ab104139). Assessment of IF reaction was performed using Confocal Laser Scanning Microscope Fluoview FV3000 coupled with CellSense software (Olympus).

### 4.9. Statistical Analysis

Statistical analysis was performed using Statistica 13 (StatSoft, Krakow, Poland) and Prism 5.0 (GraphPad Software, San Diego, CA, USA). The Shapiro–Wilk test was used to check whether the data presented a normal distribution. The data were compared using the non-parametric Mann–Whitney U test. When a normal distribution was found, the Student’s *t*-test was used. The Spearman’s rank correlation was used to examine the relationship between BCL11A expression and the selected factors. The Kaplan–Meier method was used to perform survival analysis. The significance of differences was determined using the log-rank test. The Cox proportional hazards model was used to examine the relationship between the factors and survival time. Based on the median, a cutoff point was determined for nuclear (negative vs. positive) and cytoplasmic BCL11A expression (0–1 vs. 2–12 points). Positive nuclear reaction involves more than 5% of stained nuclei. A *p* < 0.05 was considered statistically significant.

## Figures and Tables

**Figure 1 ijms-24-09848-f001:**
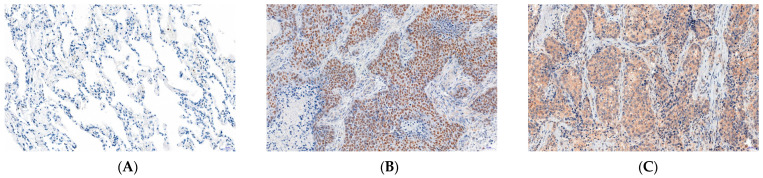
Expression of B-cell leukemia/lymphoma 11A (BCL11A) in (**A**) non-malignant lung tissue and non-small cell lung cancer: (**B**) squamous cell carcinoma (SCC) and (**C**) adenocarcinoma (AC) using immunohistochemistry. (**A**) Non-malignant lung tissue showed very low (mainly cytoplasmic) expression of BCL11A. (**B**) SCC specimens showed nuclear expression of BCL11A, while (**C**) AC samples showed cytoplasmic expression of BCL11A; magnification ×200.

**Figure 2 ijms-24-09848-f002:**
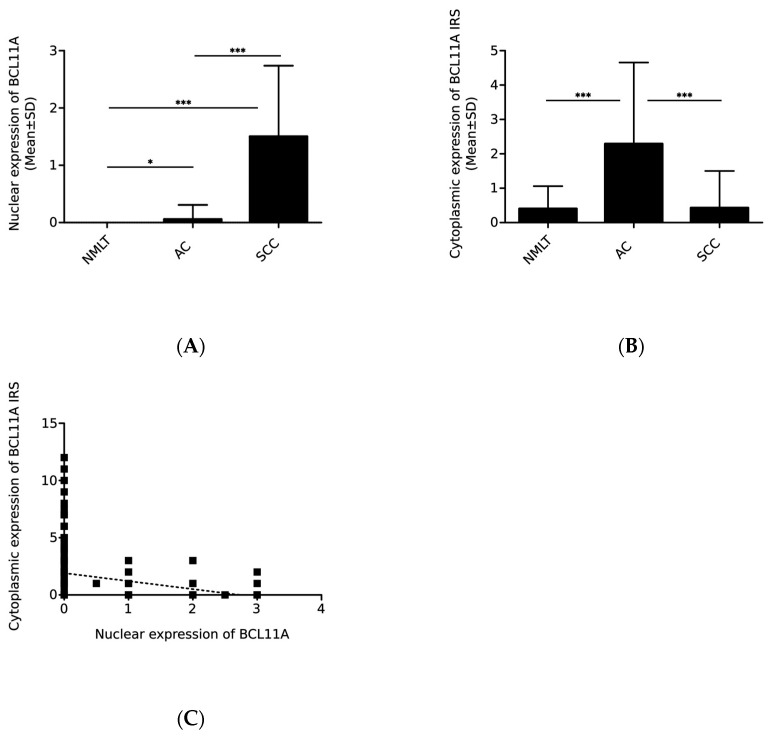
Nuclear and cytoplasmic expression levels of BCL11A in non-malignant lung tissue (NMLT) and the histological subtypes of non-small cell lung cancer (NSCLC). Comparison of (**A**) nuclear and (**B**) cytoplasmic expression of BCL11A in NMLT and NSCLC, including squamous cell lung cancer (SCC) and adenocarcinoma (AC) (*** *p* < 0.0001, * *p* = 0.016, Mann–Whitney U test). (**C**) Correlation between nuclear and cytoplasmic expression levels of BCL11A in NSCLC cells (r = −0.487, *p* < 0.0001, Spearman’s rank correlation).

**Figure 3 ijms-24-09848-f003:**
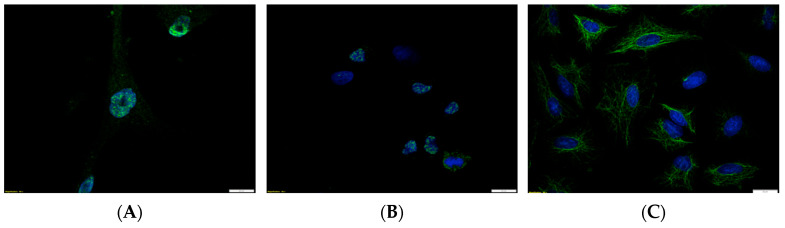
Immunofluorescence showing expression of BCL11A (green) in cell lines. Cell nuclei were visualized using DAPI dye (blue). (**A**) The IMR-90 normal lung fibroblast cell line showed low (mainly cytoplasmic) expression of BCL11A. (**B**) NCI-H1703 derived from squamous cell carcinoma (SCC) of the lung presented nuclear expression of BCL11A, while in (**C**), the A549 cell line derived from adenocarcinoma (AC) showed cytoplasmic expression of BCL11A; magnification ×600.

**Figure 4 ijms-24-09848-f004:**
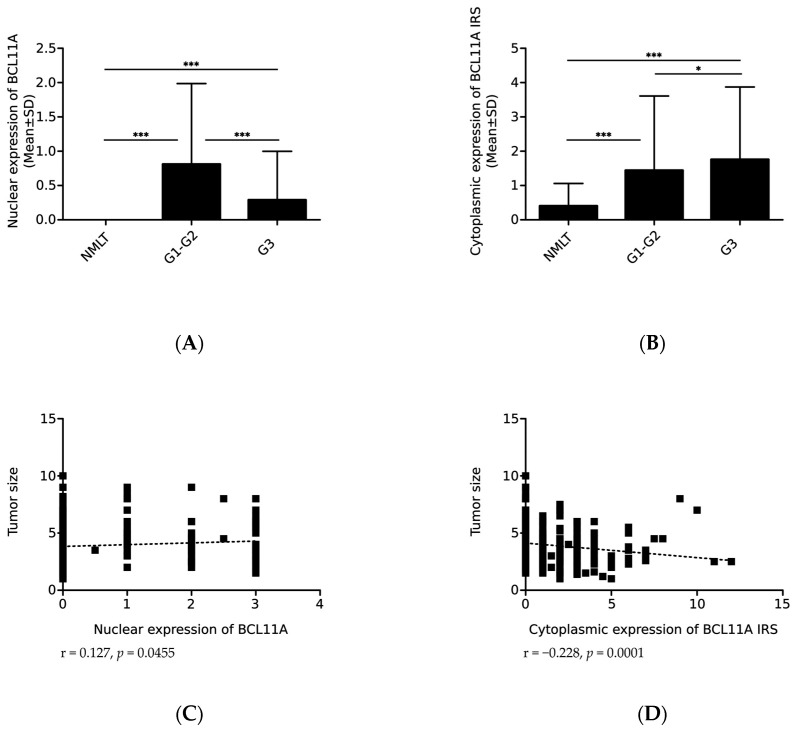
BCL11A expression levels assessed using immunohistochemistry in relation to the clinicopathological factors in non-small cell lung cancer (NSCLC). The levels of (**A**) nuclear and (**B**) cytoplasmic expression of BCL11A in non-malignant lung tissue (NMLT) and NSCLC cases depending on the histological grade (G) (*** *p* < 0.001, * *p* = 0.0325, Mann–Whitney U test). Correlation of nuclear and cytoplasmic expression of BCL11A with (**C**,**D**) tumor size in cm (r = 0.127, *p* = 0.0455; r = −0.228, *p* = 0.0001) and (**E**,**F**) necrosis (%) (r = 0.233, *p* = 0.0006; r = −0.299, *p* < 0.0001, Spearman’s rank correlation).

**Figure 5 ijms-24-09848-f005:**
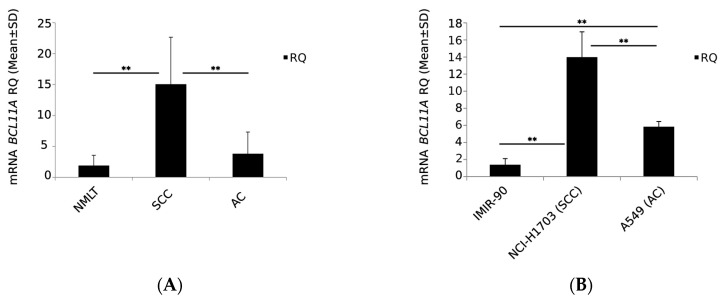
BCL11A mRNA expression levels in (**A**) non-malignant lung tissue (NMLT) and the histological subtypes of non-small cell lung cancer (NSCLC), i.e., squamous cell carcinoma (SCC) and adenocarcinoma (AC) (** *p* < 0.01, Mann–Whitney U test) and (**B**) cell lines that represent individual histological subtypes of NSCLC, i.e., NCI-H1703 (SCC), A549 (AC) and IMR-90 as control (** *p* < 0.01, Student’s *t*-test).

**Figure 6 ijms-24-09848-f006:**
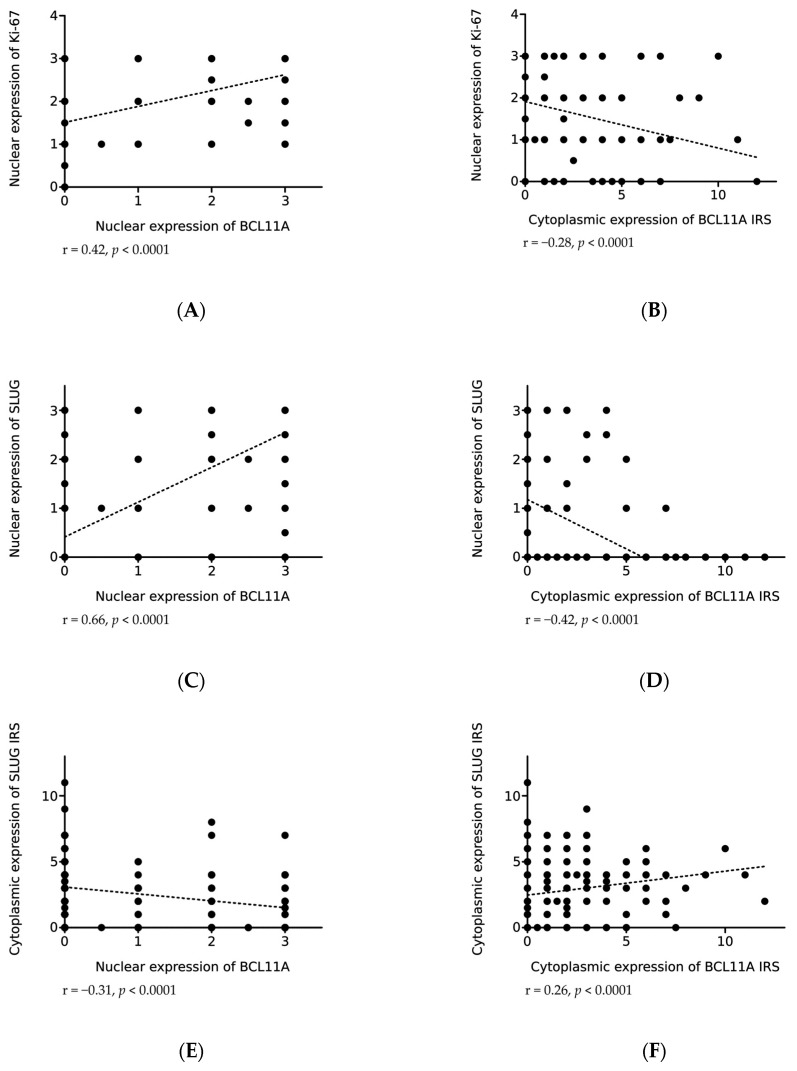
Correlations of nuclear (**A**,**C**,**E**,**G**) and cytoplasmic (**B**,**D**,**F**,**H**) expression of BCL11A with expression of Ki-67 (**A**, r = 0.42, *p* < 0.0001; **B**, r = −0.28, *p* < 0.0001), Slug (**C**, r = 0.66, *p* < 0.0001; **D**, r = −0.42, *p* < 0.0001; **E**, r = −0.31, *p* < 0.0001; **F**, r = 0.26, *p* < 0.0001) and Twist (**G**, r = 0.47, *p* < 0.0001; **H**, r = −0.21, *p* = 0.0003) in non-small cell lung cancer (NSCLC).

**Figure 7 ijms-24-09848-f007:**
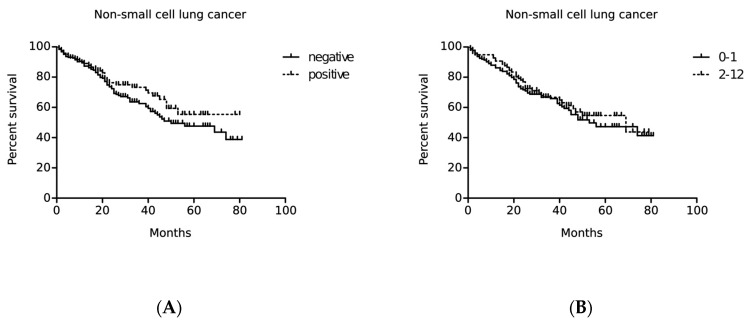
Kaplan–Meier curves showing the influence of nuclear (**A**) and cytoplasmic (**B**) expression levels of BCL11A on overall survival (OS) of patients with non-small cell lung cancer (NSCLC). The analysis was performed using the log-rank test. The cutoff point for nuclear and cytoplasmic expression of BCL11A was determined using the median. The prognostic significance of BCL11A expression was assessed in the entire cohort (**A**, *p* = 0.1850; **B**, *p* = 0.5019).

**Table 1 ijms-24-09848-t001:** Nuclear expression levels of B-cell leukemia/lymphoma 11A (BCL11A) in relation to the clinicopathological factors in non-small cell lung cancer based on immunohistochemistry.

Parameters	All Cases (*n* = 259)		SCC * (*n =* 96)		AC * (*n =* 163)	
	Mean Expression ± SD *	*p*	Mean Expression ± SD	*p*	Mean Expression ± SD	*p*
Sex						
Women	0.578 ± 1.029	0.9320	1.606 ± 1.223	0.5534	0.087 ± 0.332	0.2344
Men	0.599 ± 1.060		1.444 ± 1.251		0.032 ± 0.177	
Age						
≤65	0.508 ± 0.984	0.2356	1.488 ± 1.222	0.9509	0.024 ± 0.154	0.2132
>65	0.669 ± 1.099		1.509 ± 1.260		0.077 ± 0.313	
Histological grade						
G1, G2	0.772 ± 1.169	**0.0003**	1.506 ± 1.269	0.7913	0.047 ± 0.263	0.4116
G3	0.250 ± 0.641		1.458 ± 1.033		0.064 ± 0.247	
Primary tumor size						
pT1, pT2	0.588 ± 1.050	0.1148	1.470 ± 1.243	0.7487	0.045 ± 0.242	**0.0134**
pT3, pT4	0.810 ± 1.105		1.577 ± 1.222		0.188 ± 0.403	
Primary tumor size (cm)						
<4 cm	0.536 ± 1.033	**0.0455**	1.795 ± 1.196	**0.0422**	0.040 ± 0.244	0.2295
≥4 cm	0.679 ± 1.065		1.268 ± 1.225		0.089 ± 0.288	
Lymph node metastasis status (N)						
pN0	0.666 ± 1.109	0.6032	1.455 ± 1.275	0.6718	0.057 ± 0.279	0.6225
pN1, pN2	0.534 ± 0.965		1.554 ± 1.149		0.066 ± 0.250	
Cancer stage						
I, II	0.660 ± 1.099	0.3015	1.463 ± 1.247	0.7329	0.038 ± 0.236	**0.0341**
III, IV	0.436 ± 0.877		1.583 ± 1.240		0.116 ± 0.324	
Necrosis (%)						
<32%	0.417 ± 0.867	**0.0006**	1.129 ± 1.204	0.1512	0.077 ± 0.269	0.2026
≥32%	1.036 ± 1.253		1.593 ± 1.248		0.033 ± 0.183	

* SCC—squamous cell carcinoma; AC—adenocarcinoma; SD—standard deviation. Significant *p*-values are given in bold.

**Table 2 ijms-24-09848-t002:** Cytoplasmic expression levels of BCL11A in relation to the clinicopathological factors in non-small cell lung cancer based on immunohistochemistry.

Parameters	All Cases (*n* = 259)		SCC (*n =* 96)		AC (*n =* 163)	
	Mean Expression ± SD	*p*	Mean Expression ± SD	*p*	Mean Expression ± SD	*p*
Sex						
Women	1.735 ± 2.219	0.3029	0.424 ± 0.902	0.6129	2.362 ± 2.387	0.7677
Men	1.513 ± 2.156		0.429 ± 1.160		2.239 ± 2.360	
Age						
≤65	1.627 ± 2.099	0.5893	0.293 ± 0.750	0.4062	2.325 ± 2.234	0.6738
>65	1.575 ± 2.261		0.527 ± 1.260		2.314 ± 2.512	
Histological grade						
G1, G2	1.470 ± 2.197	**0.0325**	0.464 ± 1.124	0.2775	2.465 ± 2.528	0.5080
G3	1.844 ± 2.137		0.167 ± 0.577		2.103 ± 2.173	
Primary tumor size						
pT1, pT2	1.609 ± 2.177	0.3766	0.415 ± 1.088	0.7379	2.346 ± 2.351	0.4415
pT3, pT4	1.500 ± 2.442		0.538 ± 1.050		2.281 ± 2.966	
Primary tumor size (cm)						
<4 cm	1.909 ± 2.270	**0.0001**	0.385 ± 1.290	0.7067	2.510 ± 2.296	**0.0501**
≥4 cm	1.214 ± 2.025		0.464 ± 0.914		1.964 ± 2.510	
Lymph node metastasis status (N)						
pN0	1.562 ± 2.218	0.7333	0.522 ± 1.235	0.4914	2.362 ± 2.468	0.7861
pN1, pN2	1.612 ± 2.187		0.214 ± 0.499		2.254 ± 2.360	
Cancer stage						
I, II	1.559 ± 2.208	0.4134	0.439 ± 1.134	0.7710	2.425 ± 2.441	0.4383
III, IV	1.736 ± 2.230		0.333 ± 0.651		2.128 ± 2.358	
Necrosis (%)						
<32%	1.880 ± 2.371	**<0.0001**	0.516 ± 0.962	0.2454	2.531 ± 2.565	**0.0246**
≥32%	0.637 ± 1.243		0.426 ± 1.222		1.017 ± 1.207	

**Table 3 ijms-24-09848-t003:** Univariate and multivariable Cox proportional hazard analysis of the whole group of patients with non-small cell lung cancer.

Clinicopathological Parameter	Univariate Cox Analysis		Multivariate Cox Analysis	
	HR * (95% CI *)	*p*	HR (95% CI)	*p*
Sex (men vs. women)	0.931 (0.688–1.259)	0.6432		
Age (≤65 vs. >65)	1.060 (0.789–1.425)	0.6990		
Histological subtype (AC vs. SCC)	0.663 (0.456–0.965)	**0.0317**	0.935 (0.603–1.450)	0.7633
Histological grade (G1-2 vs. G3)	0.869 (0.674–1.120)	0.2782		
Primary tumor size (pT1-T2 vs. pT3-T4)	2.348 (1.594–3.459)	**<0.0001**	2.055 (1.356–3.114)	**0.0007**
Lymph node metastasis status (N) (pN0 vs. pN1-2)	2.115 (1.534–2.916)	**<0.0001**	1.743 (1.151–2.638)	**0.0086**
Cancer stage (I-II vs. III-IV)	2.220 (1.596–3.088)	**<0.0001**	1.280 (0.818–2.005)	0.2799
BCL11A level in the nucleus (low vs. high) 0 vs. 1–3	0.631 (0.410–0.970)	**0.0360**	0.630 (0.383–1.035)	0.0682
BCL11A level in the cytoplasm (low vs. high) ≤1 vs. >1	0.910 (0.618–1.341)	0.6339		

* HR—hazard ratio; CI—confidence interval.

**Table 4 ijms-24-09848-t004:** Univariate and multivariable Cox proportional hazard analysis of the whole group of patients with squamous cell carcinoma.

Clinicopathological Parameter	Univariate Cox Analysis		Multivariate Cox Analysis	
	HR (95% CI)	*p*	HR (95% CI)	*p*
Sex (men vs. women)	0.914 (0.471–1.771)	0.7896		
Age (≤65 vs. >65)	1.590 (0.837–3.020)	0.1569		
Histological grade (G1-2 vs. G3)	1.172 (0.563–2.439)	0.6716		
Primary tumor size (pT1-T2 vs. pT3-T4)	3.645 (1.802–7.372)	**0.0003**	3.392 (1.660–6.932)	**0.0008**
Lymph node metastasis status (N) (pN0 vs. pN1-2)	1.600 (0.813–3.150)	0.1740		
Cancer stage (I-II vs. III-IV)	2.285 (1.032–5.060)	**0.0416**	1.958 (0.870–4.410)	0.1047
BCL11A level in the nucleus (low vs. high) 0 vs. 1–3	1.057 (0.487–2.294)	0.8878		
BCL11A level in the cytoplasm (low vs. high) ≤1 vs. >1	0.412 (0.090–1.877)	0.2518		

**Table 5 ijms-24-09848-t005:** Univariate and multivariable Cox proportional hazard analysis of the whole group of patients with adenocarcinoma.

Clinicopathological Parameter	Univariate Cox Analysis		Multivariate Cox Analysis	
	HR (95% CI)	*p*	HR (95% CI)	*p*
Sex (men vs. women)	1.155 (0.749–1.780)	0.5146		
Age (≤65 vs. >65)	0.819 (0.528–1.269)	0.3715		
Histological grade (G1-2 vs. G3)	1.320 (0.771–2.257)	0.3114		
Primary tumor size (pT1-T2 vs. pT3-T4)	1.338 (0.702–2.548)	0.3759		
Lymph node metastasis status (N) (pN0 vs. pN1-2)	2.519 (1.604–3.956)	**<0.0001**	2.056 (1.137–3.719)	**0.0171**
Cancer stage (I-II vs. III-IV)	2.273 (1.445–3.575)	**0.0004**	1.389 (0.767–2.515)	0.2788
BCL11A level in the nucleus (low vs. high) 0 vs. 1–3	0.985 (0.357–2.712)	0.9760		
BCL11A level in the cytoplasm (low vs. high) ≤1 vs. >1	0.851 (0.530–1.368)	0.5062		

**Table 6 ijms-24-09848-t006:** Clinicopathological data of patients whose material was used for immunohistochemical reactions.

Parameters	All Cases		AC		SCC	
	*n* = 259	100%	*n =* 163	62.9%	*n =* 96	37.1%
Age						
Range	50–82		50–82		52–82	
Mean	66.15 ± 8.32		65.43 ± 8.31		67.35 ± 8.32	
Sex						
Men	157	60.6%	94	57.7%	63	65.6%
Women	102	39.4%	69	42.3%	33	34.4%
Histological grade						
G1	2	0.8%	2	1.2%	0	0.0%
G2	167	64.5%	83	50.9%	84	87.5%
G3	90	34.7%	78	47.9%	12	12.5%
Primary tumor size						
pT1	85	32.8%	63	38.7%	22	22.9%
pT2	130	50.2%	70	42.9%	60	62.5%
pT3	24	9.3%	12	7.4%	12	12.5%
pT4	5	1.9%	4	2.5%	1	1.0%
No data	15	5.8%	14	8.6%	1	1.0%
Primary tumor size (cm)						
<4 cm	138	53.3%	99	60.7%	39	40.6%
≥4 cm	112	43.2%	56	34.4%	56	58.3%
No data	9	3.5%	8	4.9%	1	1.0%
Lymph node metastasis status (N)						
pN0	154	59.5%	87	53.4%	67	69.8%
pN1	42	16.2%	25	15.3%	17	17.7%
pN2	47	18.1%	36	22.1%	11	11.5%
No data	16	6.2%	15	9.2%	1	1.0%
Cancer stage						
I	111	42.9%	68	41.7%	43	44.8%
II	77	29.7%	38	23.3%	39	40.6%
III	52	20.1%	40	24.5%	12	12.5%
IV	3	1.2%	3	1.8%	0	0.0%
No data	16	6.2%	14	8.6%	2	2.1%
Necrosis (%)						
<32%	96	37.1%	65	39.9%	31	32.3%
≥32%	84	32.4%	30	18.4%	54	56.3%
No data	79	30.5%	68	41.7%	11	11.5%
Survival						
Survivors	153	59.1%	91	55.8%	62	64.6%
Deceased	104	40.2%	71	43.6%	33	34.4%
No data	2	0.8%	1	0.6%	1	1.0%

**Table 7 ijms-24-09848-t007:** Clinicopathological data of patients whose material was used for molecular tests.

Parameters	All Cases		AC		SCC	
	*n =* 33	100%	*n =* 12	36.4%	*n =* 21	63.6%
Age						
Range	52–77		53–77		52–77	
Mean	65.39 ± 6.85		64.28 ± 8.35		66.10 ± 6.25	
Sex						
Men	18	54.5%	5	41.7%	13	61.9%
Women	15	45.5%	7	58.3%	8	38.1%
Histological grade						
G1	0	0.0%	0	0.0%	0	0.0%
G2	19	57.6%	4	33.3%	15	71.4%
G3	14	42.4%	8	66.7%	6	28.6%
Primary tumor size						
pT1	10	30.3%	5	41.7%	5	23.8%
pT2	18	54.5%	6	50%	12	57.1%
pT3	4	12.1%	1	8.3%	3	14.3%
pT4	1	3.0%	0	0.0%	1	4.8%
Primary tumor size (cm)						
<4 cm	19	57.6%	9	75%	10	47.6%
≥4 cm	13	39.4%	2	16.7%	11	52.4%
No data	1	3.0%	1	8.3%	0	0.0%
Lymph node metastasis status (N)						
pN0	21	63.6%	7	58.3%	14	66.7%
pN1	5	15.2%	1	8.3%	4	19.0%
pN2	7	21.2%	4	33.3%	3	14.3%
Cancer stage						
I	14	42.4%	6	50%	8	38.1%
II	12	36.4%	2	16.7%	10	47.6%
III	7	21.2%	4	33.3%	3	14.3%
IV	0	0.0%	0	0.0%	0	0.0%
Necrosis (%)						
<32%	11	33.3%	7	58.3%	4	19.0%
≥32%	12	36.4%	1	8.3%	11	52.4%
No data	10	30.3%	4	33.3%	6	28.6%
Survival						
Survivors	19	57.6%	4	33.3%	15	71.4%
Deceased	13	39.4%	8	66.7%	5	23.8%
No data	1	3.0%	0	0.0%	1	4.8%

**Table 8 ijms-24-09848-t008:** Semi-quantitative immunoreactive score (IRS) according to Remmele and Stegner [38]. The final score is the product of factors A and B ranging from 0 to 12.

Factor A		Factor B	
Points	Percentage of Positive Cells	Points	Reaction Intensity
0	0%	0	no reaction
1	≤10%	1	low
2	11–50%	2	medium
3	51–80%	3	strong
4	>80%		

**Table 9 ijms-24-09848-t009:** A point scale for assessment of nuclear reaction [39].

Points	Percentage of Positive Cells
0	≤5%
1	6–25%
2	26–50%
3	>50%

## Data Availability

The raw data and the analytic methods will be made available to other researchers for the purposes of reproducing the results in their own laboratories upon reasonable request. To access protocols or datasets, contact agnieszka.gomulkiewicz@umw.edu.pl.

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
