# Peer review of "BCL11A Expression in Non-Small Cell Lung Cancers"

_ijms, 2023, doi:10.3390/ijms24129848_

Round 1

Reviewer 1 Report

The authors presented here their experience on BCL11A expression in NSCLC (259 cases) and normal lung tissue (116 samples) using IHC and IF on FFPE specimens (and cell lines. Also other markers of EMT or dismal prognosis have been included., such as slug, snali, twist and Ki67.

At the end, BCL11A expression was noted either in squamous (SCC) and adenocarcinoma (ADC) histology, but nuclear expression was higher in SCC and cytoplasmic stain in ADC.  Some staining was noted even in normal lung tissue, although on significantly lower level.

I have some criticisms about the study here, as follows:

1. The finding of nuclear and cytoplasmic staining is very unusual, and possibly due to technical problems with the clones used in the study (RUO with limited info in literature). May the authors better explain this discrepancy, generally not very common in routine practice with IHC (i.e., beta-catenin nuclear staining due to gene mutations).

2. Figure 1B seems more consistent with small cell carcinoma. I wonder if there is some troubles with diagnosis of lung cancers included in the series. Have the authors performed diagnostic IHC in poorly-differentiated carcinomas ? e.g., TTF1, p40, NE markers ?

3. The authors seem to have included oncly SCC and ADC. No NE tumors were included in the study. Is there a specific reason ?

4. It is unclear whether normal tissue is from parenchyma adjacent to lung cancer or rather from other cases with non-neoplastic conditions, such as pneumothorax. Please, state

5. The authors used the previous 2015 WHO classification to diagnose lung cancer, but there is a new 2021 edition. Please, state and correct

6. What is the cut-off on IHC to quote a case as positive ? Any staining ? Intensity or percentage ? Combination of both features ? Please, add a comment, since this finding is crucial in case of comparison with other stusied on BCL11A

Good quality 

Reviewer 2 Report

interesting article, i have no comments.

i think it could be accepted in current form

Reviewer 3 Report

I am grateful to the editor for the opportunity to review the manuscript entitled "BCL11A Expression in Non-Small Cell Lung Cancers" by KÄ…tnik et al., currently under consideration for publication in the International Journal of Molecular Sciences. The authors should be commended for their scholarly contribution to the field, presenting a comprehensive investigation into the role of B-cell lymphoma/leukaemia 11A (BCL11A) in non-small cell lung cancer (NSCLC).

The manuscript delves into the expression of BCL11A in NSCLC and its correlation with clinicopathological parameters and patient survival. The authors have meticulously examined BCL11A expression in two NSCLC cell lines and a normal lung fibroblast cell line. They have also performed tissue microarrays (TMAs) on patient samples, selecting representative spots from archival paraffin blocks based on previously prepared slides stained with haematoxylin and eosin. The specimens were scanned using a histology scanner. Immunohistochemical reactions were performed on the TMA sections, using primary antibodies directed against BCL11A and other proteins. The intensity of BCL11A expression was evaluated by two independent investigators using a light microscope. The cytoplasmic reaction was determined using a semi-quantitative immunoreactive score (IRS). The study also includes statistical analysis of the correlation between BCL11A expression and various clinicopathological parameters such as histological subtype, grade, primary tumour size, lymph node metastasis status, and cancer stage.

The manuscript's strengths lie in its thoroughness and the depth of the research conducted. The authors have employed a variety of methods to investigate the role of BCL11A in NSCLC, providing a comprehensive view of the topic. The results are presented clearly and are well-supported by the data. The findings could contribute to the understanding of the molecular mechanisms underlying NSCLC and potentially inform future therapeutic strategies.

Despite these strengths, there are areas where the manuscript could be enhanced. I offer the following suggestions for the authors' consideration:

1.      The introduction, while providing a good overview of the topic, could be enriched by a more extensive literature review. This would help to contextualize the study within the broader landscape of existing research on BCL11A and NSCLC and highlight the unique contributions of their study more effectively.

2.      The methods section, while comprehensive, could benefit from additional detail. For instance, the authors could provide more information about the selection criteria for the NSCLC cases and the controls. This would help readers to better understand the study design and assess the validity of the results.

3.      The presentation of the results could be improved by including more visual aids such as graphs and charts. This would make the data easier to interpret and would allow readers to quickly grasp the key findings of the study.

4.      Acknowledging the limitations of the study can strengthen a manuscript by demonstrating the authors' critical thinking and awareness of the nuances of their research. The authors could discuss potential limitations of their study, such as the sample size, the methods used, or any assumptions made in their analysis.

5.      The authors may consider elaborating more on the implications of their findings for both clinical practice and future research. How might their findings influence the treatment of NSCLC? What new research questions do their findings raise? This would help to underscore the significance of their work.

In conclusion, I would like to reiterate my appreciation to both the editor and the authors for the opportunity to review this manuscript. I believe that the suggested modifications, if addressed, will further improve the quality and impact of the work. I look forward to seeing the revised version and wish the authors success in their ongoing research endeavours. 

The manuscript could benefit from careful proofreading and editing to correct any grammatical errors or awkward phrasing. This would improve the readability of the manuscript and ensure that the authors' findings are communicated clearly.

Round 2

Reviewer 1 Report

None

Reviewer 3 Report

I am deeply appreciative of the diligence and thoughtfulness you have demonstrated in considering my recommendations and subsequently refining your manuscript. Your efforts in this regard are indeed commendable.

Only minor editing of English language is required.